# Does radiofrequency ablation procedural data improve the accuracy of identifying atrial fibrillation recurrence?

Mingkai Peng[1], Amit Doshi[2], Yariv Amos[3], Liat Tsoref[3], Mati Amit[3], Don Yungher[3], Rahul Khanna[1], Paul M. Coplan[1,4] *

1 Epidemiology & Real-World Data Sciences, MedTech, Johnson & Johnson, New Brunswick, New Jersey, United States of America, 2 Mercy Hospital, St. Louis, Missouri, United States of America, 3 Biosense Webster LTD, Haifa Technology Center, Haifa, Israel, 4 Perelman School of Medicine, University of Pennsylvania, Philadelphia, PA, United States of America

* pcoplan@yahoo.com

**Data Availability Statement:** The data used for this study comes from Mercy Health data. Mercy retains all right and interest of the data. Our data use agreement does not allow us to share the data

## Abstract

Radiofrequency ablation (RFA) using the CARTO 3D mapping system is a common approach for pulmonary vein isolation to treat atrial fibrillation (AF). Linkage between CARTO procedural data and patients' electronical health records (EHR) provides an opportunity to identify the ablation-related parameters that would predict AF recurrence. The objective of this study is to assess the incremental accuracy of RFA procedural data to predict post-ablation AF recurrence using machine learning model. Procedural data generated during RFA procedure were downloaded from CARTONET and linked to deidentified Mercy Health EHR data. Data were divided into train (70%) and test (30%) data for model development and validation. Automate machine learning (AutoML) was used to predict 1 year AF recurrence, defined as a composite of repeat ablation, electrical cardioversion, and AF hospitalization. At first, AutoML model only included Patients' demographic and clinical characteristics. Second, an AutoML model with procedural variables and demographical/clinical variables was developed. Area under receiver operating characteristic curve (AUROC) and net reclassification improvement (NRI) were used to compare model performances using test data. Among 306 patients, 67 (21.9%) patients experienced 1-year AF recurrence. AUROC increased from 0.66 to 0.78 after adding procedural data in the AutoML model based on test data. For patients with AF recurrence, NRI was 32% for model with procedural data. Nine of 10 important predictive features were CARTO procedural data. From CARTO procedural data, patients with lower contact force in right inferior site, long ablation duration, and low number of left inferior and right roof lesions had a higher risk of AF recurrence. Patients with persistent AF were more likely to have AF recurrence. The machine learning model with procedural data better predicted 1-year AF recurrence than the model without procedural data. The model could be used for identification of patients with high risk of AF recurrence post ablation.

without the permission from Mercy Health, Kim Collison Farr (Kimberly.CollisonFarr@Mercy.Net).

**Funding:** This study is supported by Johnson and Johnson, New Brunswick, NJ, USA without specific grant number.

**Competing interests:** MP, YA, LT, MA, DY, RK, and PC are Johnson & Johnson employees. AD has no conflicts of interest to declare.

## Introduction

Atrial fibrillation is the most common arrhythmia diagnosed in the clinical practice and is associated with increased risk of stroke, heart failure, dementia, and overall mortality [1]. Catheter ablation has been established as an effective treatment for paroxysmal and persistent AF. However, AF recurrence still occurs for around 20% to 40% of patients post ablation [2].

During ablation procedures, circumferential lesion sets around right and left pulmonary veins and additional ablation lesion lines are created to achieve the isolation of AF triggering PV foci, elimination of non-PV triggering foci, and/or as the result of modification of the arrhythmogenic substrate. Ablation factors such as catheter tip-tissue contact, power, time of ablation, catheter stability, inter-lesion distance affect lesion quality, leading to varying rates of AF recurrence [3]. Durable pulmonary vein isolation (PVI) is associated with a lower risk of AF recurrence based on meta-analysis [4]. Ablation index is a composite index of contact force, ablation time, and power. Ablation index guided ablation strategy has been shown to achieve high rate of durable PVI and freedom from arrhythmia in persistent AF patients [5]. Collection of procedural data related to lesion formation during the ablation process could improve understanding and support achievement of ablation success rates.

In this study, we linked the ablation procedural data from CARTO systems with health system EHR data. Machine learning model was developed to assess the incremental accuracy of procedural data in identifying patients with high risk of AF recurrence.

## Method

### Data sources

This study used the linked EHR data collected using EPIC software with procedural data collected from CARTO systems (Biosense Webster Inc, Irvine, CA) and stored in CARTONET cloud-based platform system during RFA procedure for AF patients in Mercy healthcare system. All CARTONET data were scrubbed of protected health information and transmitted via the Siemens teamplay platform (Siemens, Malvern, PA) to the Microsoft Azure cloud (Microsoft, Redmond, WA). Anonymized procedural data were processed using the CARTONET cloud-based analytical software. CARTONET is the cloud-based platform used to store, analyze, and share patients' case data from CARTO system. Detailed information on how the CARTONET collect, store, and analyze the data can be found over here [6].

### Study population

Mercy healthcare system operates in four states in the Midwest with over 40 hospitals, 12 outpatient surgery centers and 35 urgent care sites, and provides care to approximately 4.2 million patients. Patients (19 years and older) with AF (International Classification of Diseases, 10th revision (ICD-10), Clinical Modification (ICD-10-CM) diagnostic code I48.X) undergoing catheter ablation (identified using ICD-10-procedure codes (ICD-10-PCS) (ICD-10-PCS codes: 02563ZZ, 02573ZZ, 025K3ZZ, 025L3ZZ, 02583ZZ, 02553ZZ, 025M3ZZ, 025S3ZZ, 025T3ZZ) /current procedural terminology (CPT) (CPT code: 93656)) using either the THERMOCOOL SMARTTOUCH® catheter or THERMOCOOL SMARTTOUCH® SF ablation catheter (Biosense Webster Inc, Irvine, CA) in an inpatient or outpatient setting between January 1, 2016 and December 31, 2018 were included. The first occurrence of cardiac ablation procedure was considered as index ablation. Patients needed to be continuously enrolled in Mercy health for at least 12 months pre-index and 12 months post-index period to be included. Patients with catheter ablation procedure for AF within 6 months prior to the index date were excluded.

## Study outcome

The outcome of the study included 1-year AF recurrence defined as a composite of repeat ablation, AF related hospitalization, and/or direct current cardioversions (DCCV) post-index ablation.

## CARTONET procedural data

Catheter ablations involve the pulmonary vein isolation in right and left wide antral circumferential ablation (WACA) with potential additional ablation lines in right and left carina, roof line, mitral line, inferior line, posterior line, anterior line as shown in Fig 1 [7]. Each ablation lesion was automatically classified into an anatomical line or site using a validated ML algorithm with correction by a human observer as necessary. Right WACA includes lesions from right roof, right posterior, right inferior, and right anterior while left WACA include lesions from left roof, left posterior, left inferior, left anterior, and ridge. Additional ablation lines, such as right and left carina line, anterior line, roof line, posterior line, were performed as

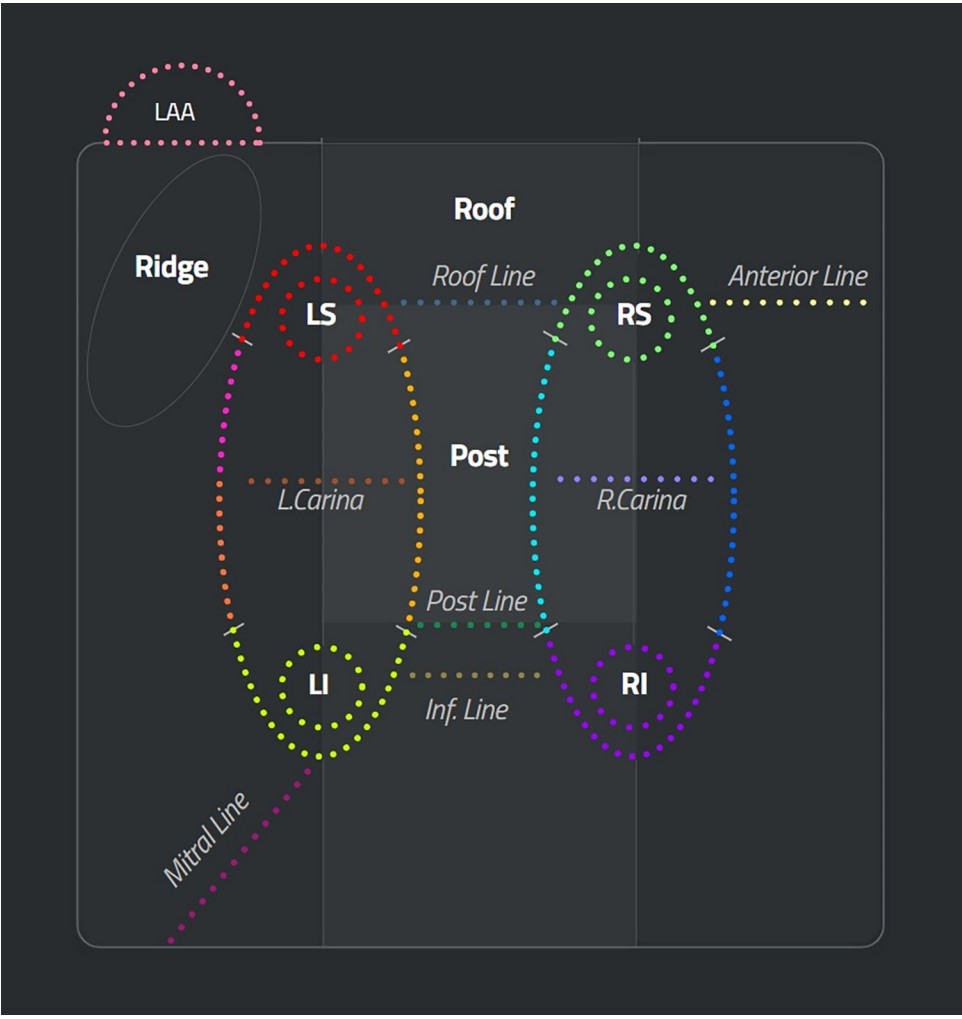

**Fig 1. Schematic illustration of ablation sites and lines on CARTONET.** Right WACA includes right posterior (light blue), right inferior (purple), right anterior (blue) and right roof (green); left WACA includes left posterior (light orange), left inferior (yellow), left anterior (orange), ridge (light purle) and left roof (red). LAA: left atrial appendage; LS: left superior pulmonary vein (PV); LI: left inferior PV; RS: right superior PV; RI: right inferior PV.

needed. The number of ablation lesions and duration of ablation were calculated for each ablation line or site. Averages of the following ablation parameters including stability [mm], average force [gr], and maximum power [W] were derived from the Visitag module for each ablation site/line. Also, total number of ablation lesions and total ablation, mapping, and procedural duration were calculated for each patient.

Ablations were not conducted at all anatomical sites or lines for every patient. If there was no ablation conducted on the anatomical lines, procedural data for the corresponding anatomical lines were set to zero before analysis.

## Statistical analysis

Demographic, clinical, and ablation characteristics were compared for patients with and without AF recurrence. Chi-square test and count/proportion were used for categorical variables. For continuous variables, t-test and mean/standard deviation (SD) were used.

AutoGluon-Tabular, an open-source automate machine learning (AutoML) framework in Python, were used to train AutoML model by ensembling multiple machine learning models (e.g. random forest, neural network, boosted tree, catBoost) and stacking them in multiple layers [8]. The dataset was divided into train data (70%) for model development and test data (30%) for model validation. At first, AutoML model included patients' demographic and clinical variables. Second, AutoML model was developed with CARTO procedural data, as well as demographic/clinical variables. AutoGluon was run with parameters "eval_metric = 'roc_auc', presets = 'best_quality', and time_limit = 1200", leaving all other parameters were left as default. The AutoGluon function automatically selects 20% of the training data for validation and hyperparameter fine-tuning.

Model performance was assessed on test data. Area under Receiver operating characteristic (AUROC) was calculated for AutoML models without and with procedural data. Feature importance is calculated using permutation method by randomly shuffling the values of each feature in the test data and calculating the resulting decrease in AUROC [9]. The features with the largest decrease in performance are considered the most important. Patients in test data were grouped in three risk categories (tercile): low, medium, and high based on the predictive probability of AF recurrence from AutoML models without procedural data. Same cutoff values of predictive probability were used for AutoML model with procedural data. Net reclassification improvement (NRI) was calculated to compare discrimination between AutoML models without and with procedural data [10].

Partial dependence plots (PDP) and accumulated local effect (ALE) plots were used to visualize how each CARTO feature influences the AF recurrence from AutoML model using test data [9]. Both PDP and ALE plots illustrate the average probability of AF recurrence when values of a study feature are changed, and all other features keep their original values. PDP used all the patients in the test data to calculate the average prediction for a unique value for the study feature. For ALE, it first identifies a subset of patients within a defined value range of the study feature, replace the study feature value with its minimum and maximum value within the subset, then calculate the difference of prediction between the subset with minimum and maximum value. The above process is repeated for all the defined intervals for the study feature and the cumulative sum of differences were calculated for ALE plots. Compared to PDP, ALE is valid if a feature is correlated with other features.

## Results

There were 306 patients in this study with 67 (21.9%) patients having the AF recurrence within 1 year post ablation. As shown in Table 1, patients with persistent AF were more likely to have

**Table 1. Patient characteristics stratified by AF recurrence.**

| | AF recurrence | | |
| --- | --- | --- | --- |
| | No (n = 239) | Yes (n = 67) | P value |
| **Patient characteristics** | | | |
| Age, mean (SD) | 63.5 (10.2) | 65.2 (9.7) | 0.23 |
| Male, n (%) | 161 (67.4) | 47 (70.1) | 0.78 |
| AF type, n (%) | | | < 0.001 |
| Paroxysmal AF | 131 (54.8) | 17 (25.4) | |
| Persistent AF | 106 (44.4) | 49 (73.1) | |
| Unknown type | 2 (0.8) | 1 (1.5) | |
| Sleep apnea, n (%) | 72 (30.1) | 20 (29.9) | 0.99 |
| Elixhauser score, mean (SD) | 2.9 (2.2) | 3.5 (2.2) | 0.09 |
| CHA$_2$DS$_2$-VASc score, mean (SD) | 2.0 (1.6) | 2.3 (1.5) | 0.12 |

AF: atrial fibrillation; SD: standard deviation

AF recurrence than paroxysmal AF. There were no statistical differences between patients with and without AF recurrence regarding to age, gender, sleep apnea, Elixhauser score, and CHA$_2$DS$_2$-VASc score.

As shown in Table 2. patients with AF recurrence had longer ablation and mapping duration, and higher number of total ablation lesions than patients without AF recurrence. Patients with AF recurrence had higher number of lesions in left anterior, left ridge, left roof, left WACA, and total number of lesions, as well as lower right inferior force than patients without AF recurrence.

Fig 2 shows the ablation patterns of study samples. Ablation pattern is defined as the combinations of ablation status in the anatomical sites/lines. The Y axis shows the ablation sites/lines and the number of patients with ablation conducted at the corresponding sites/lines. Red tiles indicate ablation were conducted. All the patients have ablations conducted at the right and left WACA. However, there were a few patients with incomplete PVI at some anatomical sites, especially for right inferior (25 patients), and left roof (13 patients). Ablation lines were only conducted at subset of patients. The most common ablation lines were right carina (187 patients), left carina (109 patients), and roof line (141 patients). Only 25 patients had anterior line ablation and 68 patients with posterior line ablation.

After adding procedural data to the AutoML model, AUROC increased from 0.66 to 0.78 as shown on Fig 3(A). Patients in the test data were grouped into low, medium, and high-risk categories (tercile) based on predicted probability of AF recurrence from AutoML model. As shown on Fig 3(B), in the low-risk category, AutoML without procedural data had 12.9% vs 0% of AF recurrence for model with procedural data, respectively; For the high-risk category, AutoML model without procedural data had 32.2% of AF recurrence vs 44.8% of AF recurrence for model with procedural data.

NRI for patients with and without AF recurrence are 32% and 4% respectively, as shown in Table 3. For patients with AF recurrence, there were 8 (42%) patients moved to higher risk categories when adding procedural data to model. There were no patients in low-risk category in the AutoML model with procedural data. For patients without AF recurrence, more patients are in low-risk category in AutoML model with procedural data with the NRI of 4%.

Table 4 shows the top 10 important predictive features for AF recurrence from AutoML model with procedural data. Nine of 10 features were from CARTO procedural data. Right inferior force and AF type had the feature importance of 0.08. It indicates AUROC would

**Table 2. Ablation procedural characteristics stratified by AF recurrence.**

| | AF recurrence | | |
| --- | --- | --- | --- |
| | No (n = 239) | Yes (n = 67) | P value |
| Mapping duration in minutes, mean (SD) | 74.9 (33.3) | 89.6 (36.6) | 0.002 |
| Ablation duration in minutes, mean (SD) | 53.8 (22.5) | 62.3 (24.2) | 0.01 |
| Total number of lesions, mean (SD) | 172.7 (82.9) | 197.4 (95.5) | 0.04 |
| Right WACA | | | |
| Number of lesions, n (%) | 72.2 (35.5) | 72.9 (37.8) | 0.89 |
| Ablation duration, mean (SD) | 19.6 (8.9) | 19.4 (8.7) | 0.88 |
| Stability, mean (SD) | 1.6 (0.6) | 1.6 (0.5) | 0.51 |
| Right anterior | | | |
| # of patients without ablation, n (%) | 1 (0.4) | 0 (0.0) | 1 |
| # of lesions, n (%) | 28.7 (17.9) | 30.7 (18.1) | 0.42 |
| Ablation duration, mean (SD) | 18.1 (6.3) | 17.9 (6.6) | 0.82 |
| Force, mean (SD) | 16.8 (6.0) | 16.0 (6.7) | 0.33 |
| Power, mean (SD) | 29.5 (12.2) | 27.8 (12.0) | 0.31 |
| Right inferior | | | |
| # of patients without ablation, n (%) | 18 (7.5) | 7 (10.4) | 0.61 |
| # of lesions, n (%) | 6.8 (3.7) | 6.6 (3.4) | 0.79 |
| Ablation duration, mean (SD) | 18.1 (6.3) | 17.5 (6.9) | 0.52 |
| Force, mean (SD) | 17.2 (7.0) | 14.4 (5.8) | 0.01 |
| Power, mean (SD) | 24.6 (10.2) | 24.4 (9.9) | 0.86 |
| Right posterior | | | |
| # of patients without ablation, n (%) | 4 (1.7) | 0 (0.0) | 0.65 |
| # of lesions, n (%) | 23.7 (14.5) | 23.0 (14.0) | 0.72 |
| Ablation duration, mean (SD) | 15.8 (5.1) | 15.7 (5.5) | 0.89 |
| Force, mean (SD) | 14.7 (5.5) | 13.4 (4.9) | 0.08 |
| Power, mean (SD) | 22.4 (9.8) | 23.1 (10.1) | 0.59 |
| Right roof | | | |
| # of patients without ablation, n (%) | 2 (0.8) | 2 (3.0) | 0.45 |
| # of lesions, n (%) | 14.4 (9.8) | 13.8 (9.9) | 0.62 |
| Ablation duration, mean (SD) | 14.8 (6.1) | 14.9 (6.3) | 0.94 |
| Force, mean (SD) | 14.1 (4.5) | 14.9 (6.0) | 0.27 |
| Power, mean (SD) | 26.2 (10.5) | 26.6 (10.3) | 0.78 |
| Left WACA | | | |
| Number of lesions, n (%) | 61.9 (34.0) | 74.3 (45.0) | 0.02 |
| Ablation duration, mean (SD) | 16.7 (7.7) | 18.9 (9.3) | 0.05 |
| Stability, mean (SD) | 1.6 (0.6) | 1.6 (0.5) | 0.89 |
| Left anterior | | | |
| # of patients without ablation, n (%) | 1 (0.4) | 2 (3.0) | 0.24 |
| # of lesions, n (%) | 12.9 (9.7) | 17.5 (13.6) | 0.002 |
| Ablation duration, mean (SD) | 17.2 (7.6) | 15.9 (6.3) | 0.24 |
| Force, mean (SD) | 12.7 (4.9) | 12.8 (5.9) | 0.92 |
| Power, mean (SD) | 29.2 (12.1) | 28.3 (12.3) | 0.63 |
| Left inferior | | | |
| # of patients without ablation, n (%) | 4 (1.7) | 5 (7.5) | 0.04 |
| # of lesions, n (%) | 7.2 (4.8) | 7.3 (6.2) | 0.88 |
| Ablation duration, mean (SD) | 16.1 (6.1) | 15.1 (5.5) | 0.22 |
| Force, mean (SD) | 12.8 (5.0) | 12.7 (5.5) | 0.87 |

(*Continued*)

**Table 2.** (Continued)

| | AF recurrence | | |
|---|---|---|---|
| | No (n = 239) | Yes (n = 67) | P value |
| Power, mean (SD) | 23.3 (10.5) | 25.3 (10.2) | 0.19 |
| **Left posterior** | | | |
| # of patients without ablation, n (%) | 1 (0.4) | 4 (6.0) | 0.01 |
| # of lesions, n (%) | 19.5 (13.5) | 23.5 (24.6) | 0.09 |
| Ablation duration, mean (SD) | 15.3 (4.8) | 15.1 (4.9) | 0.76 |
| Force, mean (SD) | 13.7 (4.4) | 12.7 (4.9) | 0.15 |
| Power, mean (SD) | 20.6 (9.5) | 22.2 (9.3) | 0.25 |
| **Left roof** | | | |
| # of patients without ablation, n (%) | 11 (4.6) | 2 (3.0) | 0.81 |
| # of lesions, n (%) | 12.2 (9.7) | 15.0 (10.3) | 0.04 |
| Ablation duration, mean (SD) | 16.3 (6.4) | 15.9 (6.4) | 0.70 |
| Force, mean (SD) | 13.8 (5.0) | 12.9 (4.2) | 0.19 |
| Power, mean (SD) | 25.8 (11.1) | 26.0 (11.3) | 0.89 |
| **Left ridge** | | | |
| # of patients without ablation, n (%) | 5 (2.1) | 1 (1.5) | 1 |
| # of lesions, n (%) | 14.1 (11.4) | 18.5 (12.0) | 0.01 |
| Ablation duration, mean (SD) | 18.5 (7.4) | 17.0 (6.5) | 0.145 |
| Force, mean (SD) | 12.6 (3.9) | 12.8 (5.0) | 0.72 |
| Power, mean (SD) | 29.3 (12.4) | 28.9 (12.4) | 0.85 |
| **Right carina** | | | |
| # of patients without ablation, n (%) | 102 (42.7) | 17 (25.4) | 0.02 |
| Number of lesions, n (%) | 8.1 (7.9) | 9.3 (8.0) | 0.35 |
| Ablation duration, mean (SD) | 1.9 (1.9) | 2.4 (2.2) | 0.18 |
| Stability, mean (SD) | 1.7 (0.7) | 1.6 (0.6) | 0.55 |
| **Left carina** | | | |
| # of patients without ablation, n (%) | 163 (68.2) | 34 (50.7) | 0.01 |
| Number of lesions, n (%) | 4.7 (5.3) | 4.9 (3.6) | 0.82 |
| Ablation duration, mean (SD) | 1.1 (1.5) | 1.2 (0.9) | 0.80 |
| Stability, mean (SD) | 1.9 (0.9) | 1.8 (0.6) | 0.65 |
| **Anterior line** | | | |
| # of patients without ablation, n (%) | 224 (93.7) | 57 (85.1) | 0.04 |
| Number of lesions, n (%) | 9.1 (9.3) | 10.0 (8.2) | 0.80 |
| Ablation duration, mean (SD) | 2.8 (2.7) | 3.4 (3.4) | 0.58 |
| Stability, mean (SD) | 1.7 (0.6) | 1.6 (0.6) | 0.88 |
| **Roof line** | | | |
| # of patients without ablation, n (%), n (%) | 141 (59.0) | 24 (35.8) | 0.001 |
| Number of lesions, n (%) | 14.9 (15.4) | 17.3 (10.8) | 0.36 |
| Ablation duration, mean (SD) | 3.5 (3.4) | 4.4 (3.5) | 0.15 |
| Stability, mean (SD) | 1.9 (0.7) | 1.8 (0.6) | 0.15 |
| **Posterior line** | | | |
| # of patients without ablation, n (%) | 189 (79.1) | 49 (73.1) | 0.39 |
| Number of lesions, n (%) | 7.6 (8.6) | 6.7 (5.7) | 0.68 |
| Ablation duration, mean (SD) | 1.9 (2.2) | 1.6 (1.4) | 0.58 |
| Stability, mean (SD) | 1.9 (0.7) | 1.7 (0.7) | 0.27 |
| **Undefined** | | | |
| # of patients without ablation, n (%) | 56 (23.4) | 15 (22.4) | 0.99 |

(*Continued*)

**Table 2.** (Continued)

| | AF recurrence | | |
| --- | --- | --- | --- |
| | No (n = 239) | Yes (n = 67) | P value |
| Number of lesions, n (%) | 31.7 (26.9) | 34.1 (24.8) | 0.56 |
| Ablation duration, mean (SD) | 7.7 (6.1) | 8.6 (5.7) | 0.31 |
| Stability, mean (SD) | 1.8 (0.6) | 1.8 (0.6) | 0.93 |

WACA: wide antral circumferential ablation; SD: standard deviation

decrease by 0.08 if column of right inferior force or AF type were randomly shuffled while all other features were unchanged. The curves from PDP and ALE as shown on Figs 4 and 5 were similar for the top 9 important procedural features. Lower right inferior forces and low number of left inferior and right roof lesions are associated with high risk of AF recurrence. Long ablation duration and high number of left ridge lesions increased the risk of AF recurrence.

## Discussion

Adding CARTO procedural data to AutoML model improve the identification of patients who would experience AF recurrence. Nine of 10 top predictive features were from CARTO procedural data. AutoML model with procedural data provides important actionable insights to improve the ablation process.

Catheter ablation is a complex process that requires physicians use a diagnostic catheter to map the left atria and use a therapeutic catheter to abate cardiac tissue to block abnormal electrical signals. Successful ablation requires physicians to create lesions with appropriate contact force and power. Location of lesions is also critical as it determines whether lesions can durably block the transmission of abnormal electrical signals [7]. Automated segmentation of lesions into specific anatomical sites and calculation of their ablation parameters provide objective feedback for physicians to review their ablation process. Linking procedural data with EHR data allowed us to build machine learning model to predict AF recurrence. Our goal is to build models that can be used to inform cardiac ablation workflow and thereby improve patient outcomes.

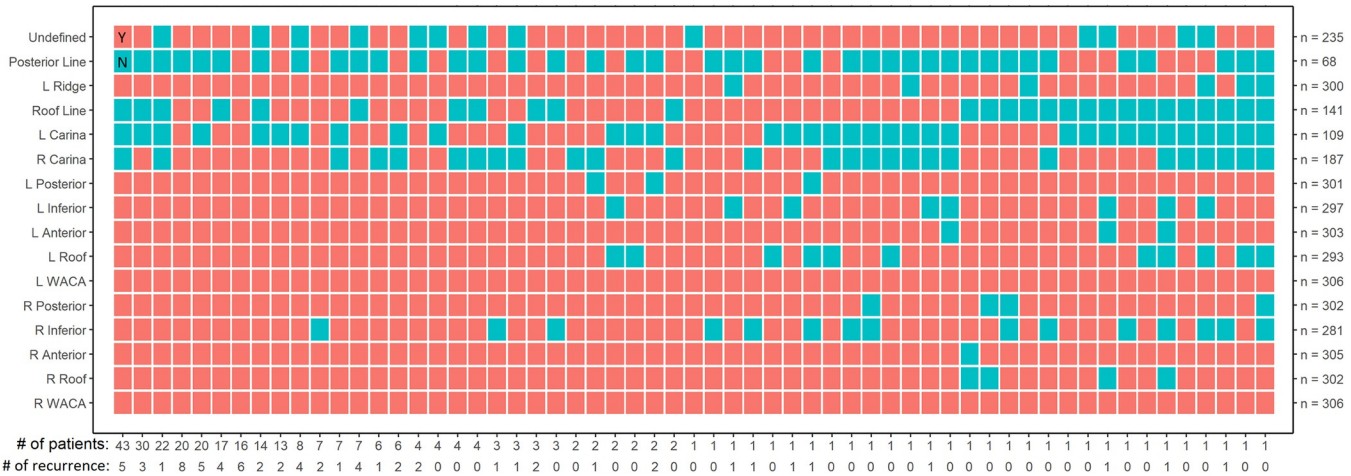

**Fig 2. Ablation patterns in the study sample shown in columns and each row shows whether ablation was conducted the specific atrial anatomical sites/lines (red tiles indicate an ablation was conducted (Y) and teal tiles indicate no ablation at that sites (N)).** Outcomes represents the number of AF recurrence among patients with the specific ablation pattern as shown in the bottom of figure.

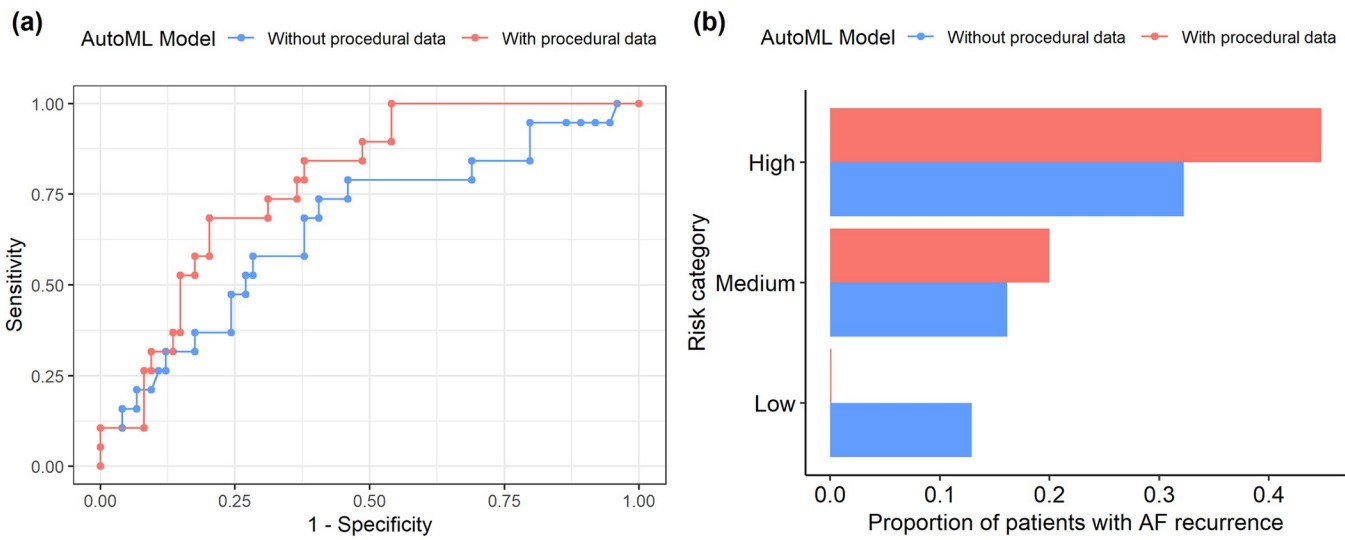

**Fig 3. Model Performance between AutoML models without and with procedural data.** (a): AUROC curves, (2) proportion of patients with AF recurrence in the risk categories.

AutoML model identified some important features that predicts the AF recurrence and provides directions for improvement in ablation strategies. Nine out of top 10 predictive features were from procedural data. Low right inferior contact force was associated with high risk of AF recurrence. Contact force provides an actionable direction for physicians to improve success rate of ablation during procedures. In this study, some patients had no ablation conducted at right inferior area. This could be due to great catheter instability around the right sided veins [11]. The ability to deliver force to right inferior atrium may be dependent on operator experience. Left inferior and right roof are from left and right WACA lines. Low number of lesions in left inferior and right roof were associated with high risk of AF recurrence. Therefore, complete PVI with more ablation lesions to completely block the abnormal electrical signals in right and left WACA have potentials to improve the success of ablation.

As shown in other studies, patients with persistent AF are more likely to have AF recurrence. One of the key features predicting AF recurrence was AF type, with persistent AF patients significantly more likely to have recurrence as compared to patients with paroxysmal

**Table 3. Reclassification of predicted risks after adding procedural data to model for patients with and without AF recurrence.**

| Risk category (Without procedural data) | Risk category (with procedural data) | | | Reclassified | | |
| --- | --- | --- | --- | --- | --- | --- |
| | **Low** | **Medium** | **High** | **Higher** | **Lower** | **NRI** |
| Patients with AF recurrence | | | | | | |
| Low | 0 | 3 | 1 | 8 (42%) | 2 (10%) | 32% |
| Medium | 0 | 1 | 4 | | | |
| High | 0 | 2 | 8 | | | |
| Patients without AF recurrence | | | | | | |
| Low | 14 | 11 | 2 | 19 (25.7%) | 22 (29.7%) | 4% |
| Medium | 11 | 9 | 6 | | | |
| High | 6 | 5 | 10 | | | |

NRI: net classification improvement

**Table 4. Top 10 important features from AutoML CARTO model.**

| Feature | Importance | Standard Deviation | P value |
|---|---|---|---|
| Right Inferior Force | 0.080 | 0.027 | 0.001 |
| AF Type | 0.079 | 0.027 | 0.001 |
| Ablation Duration | 0.048 | 0.014 | 0.001 |
| Left Inferior Ablation Lesions | 0.028 | 0.006 | 0.000 |
| Left Ridge Ablation Lesions | 0.024 | 0.027 | 0.057 |
| Right Roof Ablation Lesions | 0.014 | 0.019 | 0.085 |
| Left Roof Duration | 0.010 | 0.003 | 0.000 |
| Right Posterior Duration | 0.010 | 0.017 | 0.136 |
| Right Roof Duration | 0.008 | 0.007 | 0.036 |
| Left Inferior Duration | 0.008 | 0.008 | 0.051 |

AF. Persistent AF is the advanced stage of AF with long lasting episodes of arrhythmia than paroxysmal AF. Early ablation (referred as RFA within 1 year of AF diagnosis) can lead to improvements in adverse cardiac remodeling, greater success in maintain sinus rhythm, and decreased need for repeat ablation based on the results of clinical trials and systematic reviews [12, 13]. Based on meta-analysis results of randomized trials, catheter ablation as first-line rhythm control therapy was associated with lower incidence of atrial tachyarrhythmias

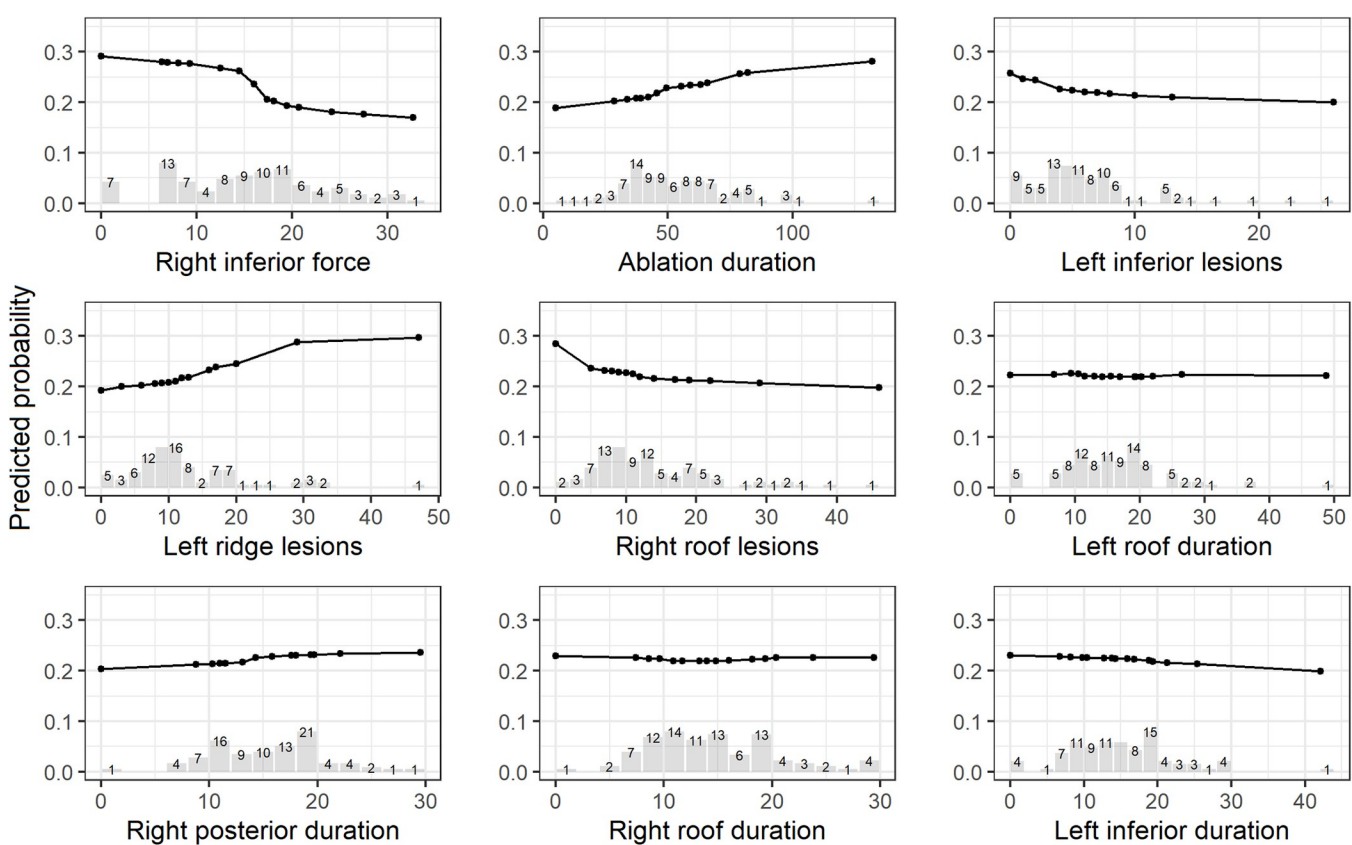

**Fig 4. Partial dependence plots of top 9 procedural features from AutoML model with procedural data (adjusted for covariates described in the Methods section).** Histogram shows the distribution of patients for each feature.

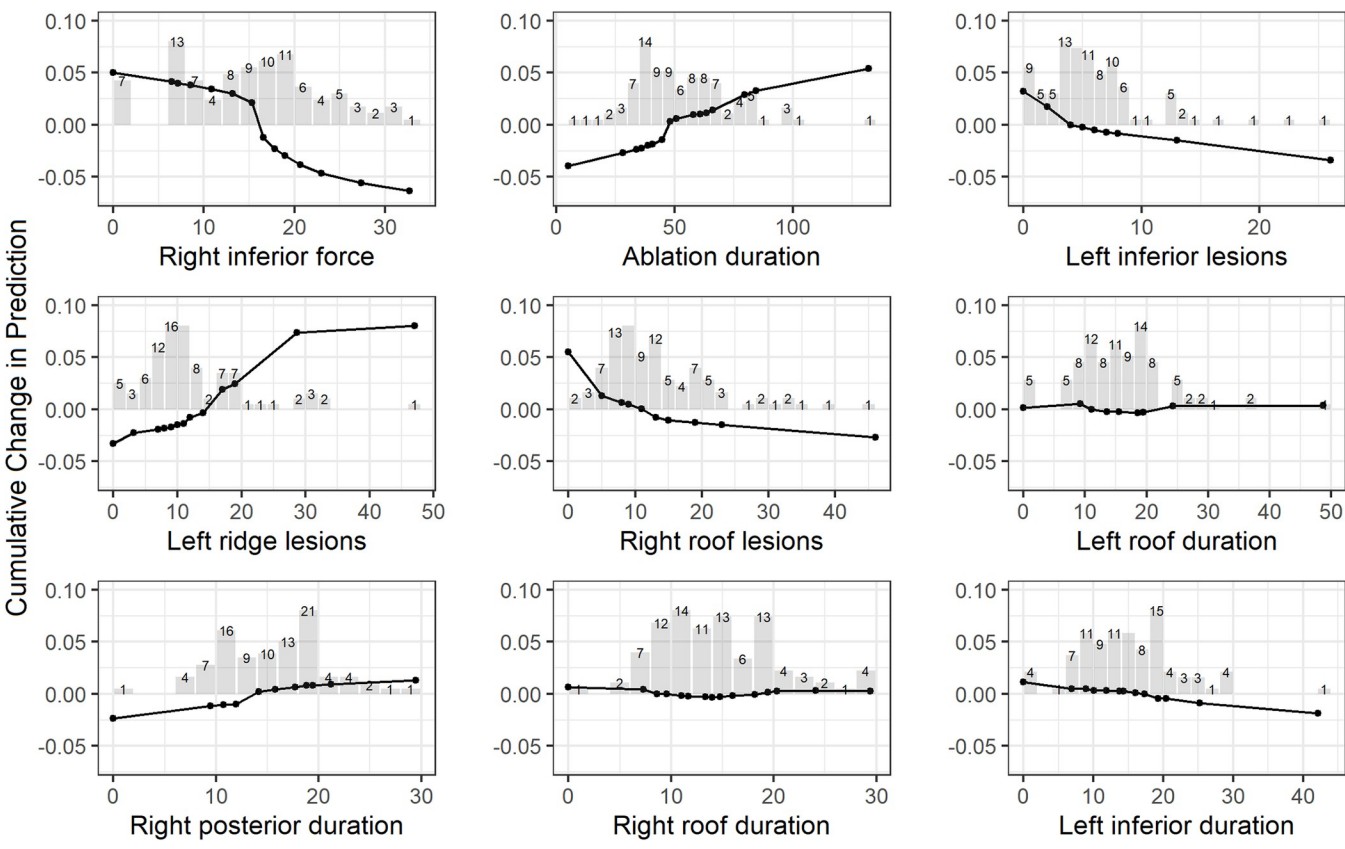

**Fig 5. Accumulated local effects plots of top 9 procedural features from AutoML model with procedural data (adjusted for covariates described in the Methods section).** Histogram shows the distribution of patients for each feature.

recurrence and hospitalizations compared with AADs, with similar incidence of death, stroke/ transient ischemic attack, and serious adverse effects [14].

Identifying patients with high risk of AF recurrence is an important step to improve the ablation success rate. Various models were developed to predict recurrent atrial fibrillation after catheter ablation [15]. For example, "FLAME score" (range 0–9) were developed for non-paroxysmal atrial fibrillation ablation and included sex, AF type, left atrial diameter, mild to moderate mitral regurgitation (MR), and extreme comorbidity [16]. Extreme comorbidity was defined as the presence of any one of severe mitral regurgitation (MR), moderate or severe mitral stenosis, mitral valve replacement, hypertrophic cardiomyopathy or structural congenital heart disease. C-statistics for FLAME score is 0.64 and patients with score >= 5 had statistically significant high risk of AF recurrence. Similarly, ATLAS score was developed to predict AF recurrence after the first catheter ablation procedures and included age, sex, smoking, AF type, left atrial volume [17]. However, both FLAME and ATLAS score focused on the pre-operative variables and did not include the intraprocedural variables. Our results demonstrate the importance of ablation procedural data in identifying patients with high risk of AF recurrence. Our model could be used to identify patients with high risk of AF recurrence post-ablation. Our model can also be integrated into the cardiac ablation technology system to provide real-time feedback to electrophysiologists during the procedure, enabling them to achieve better results. The model also identified areas for improvement in ablation procedures and provides directions to improve ablation strategies.

## Limitation

This study used the linked data between procedural data and EHR data. AF recurrence was defined using diagnosis codes and procedurals codes from EHR data. AF recurrence is more likely underestimated as recurrence are frequently asymptomatic and our AF recurrence definition are more likely to capture severe cases with required encounters with hospitals or physicians. Some procedural data, such as impedance drop, and ablation index had missing values for some patients and were excluded from AutoML model. Future model can be developed by including all ablation parameters. The study results were evaluated on the test data that were not used for model development. External validation should be conducted to use independent validation dataset to evaluate whether model would be generalized to patients outside of Mercy Health. The ablation procedural data comes from patients using Thermocool catheters and CARTO platform. External validation should be conducted to evaluate model performance on catheters and platforms from other manufacturers. Future changes of ablation parameters are anticipated as more physicians use CARTONET to review their ablation process and improve their procedures. Future updates on the model should be conducted to reflect the newest practice of ablation.

## Conclusion

AutoML model with procedural data achieved improved discrimination ability than AutoML model without procedural data. There is significant potential for CARTONET and AutoML model with procedural data to provide rich insights into factors influencing AF recurrence and thereby lead to improvements in AF ablation success rate.

## Author Contributions

**Conceptualization:** Mingkai Peng, Liat Tsoref, Paul M. Coplan.

**Data curation:** Amit Doshi, Yariv Amos, Liat Tsoref, Paul M. Coplan.

**Formal analysis:** Mingkai Peng.

**Methodology:** Mingkai Peng, Paul M. Coplan.

**Supervision:** Paul M. Coplan.

**Writing – original draft:** Mingkai Peng.

**Writing – review & editing:** Mingkai Peng, Amit Doshi, Yariv Amos, Liat Tsoref, Mati Amit, Don Yungher, Rahul Khanna, Paul M. Coplan.

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
