## [Decision Letter · Decision Letter 0]

10 Dec 2023

PONE-D-23-31440Does radiofrequency ablation procedural data improve the accuracy of identifying atrial fibrillation recurrence?PLOS ONE

Dear Dr. Peng,

Thank you for submitting your manuscript to PLOS ONE. After careful consideration, we feel that it has merit but does not fully meet PLOS ONE’s publication criteria as it currently stands. Therefore, we invite you to submit a revised version of the manuscript that addresses the points raised during the review process.

We look forward to receiving your revised manuscript.

Kind regards,

Ibrahim Marai, MD

Academic Editor

PLOS ONE

2. You indicated that ethical approval was not necessary for your study. We understand that the framework for ethical oversight requirements for studies of this type may differ depending on the setting and we would appreciate some further clarification regarding your research. Could you please provide further details on why your study is exempt from the need for approval and confirmation from your institutional review board or research ethics committee (e.g., in the form of a letter or email correspondence) that ethics review was not necessary for this study? Please include a copy of the correspondence as an ""Other"" file.

“Several authors are employed by Johnson & Johnson or Biosense Webster, a Johnson & Johnson company. However, no Johnson and Johnson or Biosense Webster product was studied in this research project.”

6. PLOS requires an ORCID iD for the corresponding author in Editorial Manager on papers submitted after December 6th, 2016. Please ensure that you have an ORCID iD and that it is validated in Editorial Manager. To do this, go to ‘Update my Information’ (in the upper left-hand corner of the main menu), and click on the Fetch/Validate link next to the ORCID field. This will take you to the ORCID site and allow you to create a new iD or authenticate a pre-existing iD in Editorial Manager. Please see the following video for instructions on linking an ORCID iD to your Editorial Manager account: https://www.youtube.com/watch?v=_xcclfuvtxQ.

7. We note that Figure 1 in your submission contain copyrighted images. All PLOS content is published under the Creative Commons Attribution License (CC BY 4.0), which means that the manuscript, images, and Supporting Information files will be freely available online, and any third party is permitted to access, download, copy, distribute, and use these materials in any way, even commercially, with proper attribution. For more information, see our copyright guidelines: http://journals.plos.org/plosone/s/licenses-and-copyright.

Reviewers' comments:

Reviewer's Responses to Questions

**Comments to the Author**

1. Is the manuscript technically sound, and do the data support the conclusions?

Reviewer #1: No

Reviewer #2: Yes

2. Has the statistical analysis been performed appropriately and rigorously? 

Reviewer #1: I Don't Know

Reviewer #2: Yes

3. Have the authors made all data underlying the findings in their manuscript fully available?

Reviewer #1: No

Reviewer #2: Yes

4. Is the manuscript presented in an intelligible fashion and written in standard English?

Reviewer #1: Yes

Reviewer #2: Yes

5. Review Comments to the Author

Reviewer #1: The study evaluates how the addition of procedural data can be used to predict risk of 12 month AF recurrence following radio frequency ablation. The study finds that in addition to AF type duration of ablation and number of lesions in specific regions recorded during the procedure improve prediction of outcome.

The study conclusions are consistent with previous studies complete PVI is important and type of AF is important in predicting outcome.

Concerns:

The use case for this predictive model was not clear. Ideally, prediction of 12 month recurrence would be used to decide to proceed with the ablation. This is the strength of ATLAS and FLAME, which should be recognized as a significant advantage of these approaches. But once the ablation has happened is it expected this could be used for informing advanced monitoring? The expected use case should be identified and commented on in the discussion.

The number of ablation cases performed in the hospital network seems to be very low. 300 cases over 3 years for 40 hospitals seems like a very small number. The study should say how many of the AF ablation patients were found overall and why only 300 were studied. There is a risk that this specific group may represent a subset and this is important to consider the results.

It was not clear why only CARTO was used for the analysis. None of the indices appeared to be proprietary, this needs to be explained more. The results would be significantly more interesting if they were extended to Abbot, Boston Scientific or Acutus systems

It would be good to train the model repeated times with different random seeds/initialization to confirm that the same features are identified.

An independent validation data set would significantly improve this study.

It was not clear why a train, test and validation set were not used.

The analysis could have been done with a cross-fold validation that would give an estimate of the uncertainty in the model AUROC. Is the difference between 0.66 and 0.78 real or spurious based on the specific learning instance.

The abstract conclusion on line 36 is not supported by the results and should probably be changed to the discussion line 218: The model….

Are the WACA statistics confounded with the other measurements, how would this impact results?

It would be useful to include LA size in the prediction as this is often available and provides good pre-procedure outcome prediction.

The large limitation is that this is only applied to a single vendor data set. This needs to be highlighted in the discussion.

Minor

The ability to deliver force to the right inferior atrium may be dependent on operator experience. It would be nice to add this to the discussion

line 214 determine => determines

Reviewer #2: Interesting data.

Main limitation is the relatively small sample size for a ML study

Main limitation is the relatively small sample size for a ML study

Main limitation is the relatively small sample size for a ML study

6. PLOS authors have the option to publish the peer review history of their article (what does this mean?). If published, this will include your full peer review and any attached files.

Reviewer #1: No

Reviewer #2: No

---

## [Author Response · Author response to Decision Letter 0]

13 Feb 2024

Reviewer #1: The study evaluates how the addition of procedural data can be used to predict risk of 12 month AF recurrence following radio frequency ablation. The study finds that in addition to AF type duration of ablation and number of lesions in specific regions recorded during the procedure improve prediction of outcome.

The study conclusions are consistent with previous studies complete PVI is important and type of AF is important in predicting outcome.

Concerns:

The use case for this predictive model was not clear. Ideally, prediction of 12 month recurrence would be used to decide to proceed with the ablation. This is the strength of ATLAS and FLAME, which should be recognized as a significant advantage of these approaches. But once the ablation has happened is it expected this could be used for informing advanced monitoring? The expected use case should be identified and commented on in the discussion.

Author response: Thanks for suggestion. As pointed out by the reviewer, the Female, Long-Lasting, Atrial diameter, Mitral, Extreme (FLAME) score was developed to predict the outcome of non-paroxysmal atrial fibrillation (AF) ablation using patients' demographic and clinical variables. The FLAME score provides clinicians with a tool to analyze individual patients' risk for ablation outcomes. Similarly, ATLAS is designed to estimate the risk of AF recurrence after the first catheter ablation procedure, incorporating patients' age, gender, AF type, left atrial volume, and smoking status. Both FLAME and ATLAS can be used to refine patient selection for ablation procedures and inform patient-physician discussions.

The success of the ablation procedure depends on patients' demographic and clinical profiles, as well as the ablation strategies (e.g., power target, lesion duration, ablation index) and techniques (e.g., contact force sensing catheter, catheter irrigation, high power short duration techniques) used during ablation procedures. 

Our tool can be integrated into the cardiac ablation technology system to provide real-time information to electrophysiologists during the procedure, enabling them to achieve better results. For instance, similar to the advanced technologies in modern automobiles that alert drivers when they deviate from their lane or approach a high-risk collision, our tool is designed to improve ablation effectiveness for electrophysiologists. We aim to develop tools that inform surgeons during cardiac ablation procedures about the probability of recurrence based on the ablation patterns used for patients. Additionally, we seek to offer suggestions for improving ablation locations, the number of lesions, contact force, impedance drop, voltage drop, etc., which collectively contribute to a lower probability of atrial fibrillation (AF) recurrence for patients.

Similar predictive models can be developed to guide electrophysiologists on reducing adverse events for patients. These models utilize variables from CARTO data, such as contact force, power, and duration used for lesion formation. Our overarching research goal is to enhance the benefit-risk balance of cardiac ablation procedures for patients by leveraging machine learning models running on cardiac ablation data. Importantly, our objective is not to replace the surgeon's clinical judgment but rather to provide valuable input to guide the radiofrequency ablation workflowIn comparison to ATLAS and FLAME, our predictive model incorporates procedural data during the ablation process to predict AF recurrence. The predictive value of procedural data was demonstrated in this study. The predictive model can inform patients and clinicians about the risk of AF recurrence after ablation. The model also identifies areas for improvement in the ablation procedure and provides directions to enhance ablation strategies.

The number of ablation cases performed in the hospital network seems to be very low. 300 cases over 3 years for 40 hospitals seems like a very small number. The study should say how many of the AF ablation patients were found overall and why only 300 were studied. There is a risk that this specific group may represent a subset and this is important to consider the results.

It was not clear why only CARTO was used for the analysis. None of the indices appeared to be proprietary, this needs to be explained more. The results would be significantly more interesting if they were extended to Abbot, Boston Scientific or Acutus systems

Author response: We agreed with the reviewer that the study would be more comprehensive and interesting if we could include patients using Abbott, Boston Scientific, or Acutus systems. However, we do not have access to a broader array of medical device data. There are two primary reasons for this limitation. First, Mercy Health primarily utilizes Medtronic and Thermocool catheters. Second, Mercy's data use agreement prohibits the sharing of data from one manufacturer to another. Consequently, we were unable to obtain procedural data from devices manufactured by other companies. As a result, only procedural data from the CARTO system are available to us. We acknowledge this limitation in our study and hope that our work will inspire other researchers to conduct similar studies using data from systems produced by different manufacturers. We have included this information in the limitations section.

This study focused on patients treated using Biosense Thermocool SmartTouch (ST) and Thermocool ST SurroundFlow (SF) and the CARTO machine with software version 6.0 or above. To link CARTONET data with Mercy Health electronic health records (EHR), nurses from Mercy Health reviewed the cases and assigned the case ID in the procedural case data. The inclusion criterions and data linkage process reduced the number of eligible cases for this study.

It would be good to train the model repeated times with different random seeds/initialization to confirm that the same features are identified.

Author response: We used the AutoGluon-Tabular framework, a type of automated machine learning (AutoML), to develop predictive models. AutoGluon ensembles multiple models and stacks them in multiple layers. AutoGluon utilizes the random forest, CatBoost, LightGBM, and Neural Net models as the base models and ensembles/stacks them in multiple layers. Bagging and boosting have already been used in the model development process. Because the steps what we have done, incremental benefits from different random seeds/initialization will be minimal. Feature importance was calculated on the test data after the model was finalized.

An independent validation data set would significantly improve this study. It was not clear why a train, test and validation set were not used. The analysis could have been done with a cross-fold validation that would give an estimate of the uncertainty in the model AUROC. Is the difference between 0.66 and 0.78 real or spurious based on the specific learning instance.

Author response: In this study, we had 306 patients, with 70% used for model training and 30% for model testing. This division allows us to accurately assess the model performance on test data with a large dataset. We used the default settings for the AutoGluon framework. The autogluon.TabularPredictor.fit() function automatically selects 20% of the training data for validation and hyperparameter fine-tuning. Our model development process is straightforward and well-defined, reducing the chance of data leakage in the model development process.

The Area Under the Receiver Operating Characteristic curve (AUROC) was calculated based on the test data after the model, both with and without procedure data, was finalized. 

We acknowledge external validation using data from other health stems should be conducted to validate the performance and generalizability of model. We add this in the limitation section. 

The abstract conclusion on line 36 is not supported by the results and should probably be changed to the discussion line 218: The model….

Author response: we revised the conclusion following suggestion: The model could be used for identification of patients with high risk of AF recurrence post ablation. Linkage between CARTO procedural data and patient EHR data is valuable to optimize ablation process and improve patient outcomes.

Are the WACA statistics confounded with the other measurements, how would this impact results?

Author response: Right WACA statistics are calculated as the sum or average of ablation parameters from lesions in the right roof, right posterior, right inferior, and right anterior. Meanwhile, left WACA statistics are calculated as the sum or average of ablation parameters from lesions in the left roof, left posterior, left inferior, left anterior, and ridge. As shown in Table 4, there are no ablation parameters from right and left WACA in the top 10 important features from the final model. Therefore, their absence has minimal impact on the results.

It would be useful to include LA size in the prediction as this is often available and provides good pre-procedure outcome prediction.

Author response: The left atrium size is not available in the output of CARTONET as it is not recorded as part of CARTO VISITAG. The CARTO VISITAG™ Module records catheter positions along with electrophysiological parameters acquired during RF applications, according to user preferences.

The large limitation is that this is only applied to a single vendor data set. This needs to be highlighted in the discussion.

Author response: The procedural data comes from CARTONET, which only stored the CARTO system data. The data is from a large health care. Unfortnatley, we can not get data from other manufactures due to data use agreement. We highlighted this limitation in the discussion. 

Minor

The ability to deliver force to the right inferior atrium may be dependent on operator experience. It would be nice to add this to the discussion. 

Author response: Thanks for the suggestion. We add that in the discussion. 

Line 214 determine => determines

Author response: revised.

Reviewer #2: Interesting data.

Main limitation is the relatively small sample size for a ML study

Author response: This is an observational study that involves linking two different data sources. Procedural data comes from CARTONET, requiring the use of Thermocool ST and STSF catheters, as well as the CARTO system during the ablation process. Nurses from the Mercy Health System review individual cases and assign patient IDs to facilitate linking CARTONET procedural data with Mercy Health electronic health records. The inclusion criteria and data linkage process reduce the number of eligible cases for this study.

We allocated 70% of patients for model training and 30% for testing data. This division enables us to accurately assess the model's performance on test data with a large dataset. Our model development process is straightforward, reducing the chance of data leakage during model development.

---

## [Editor Report · Decision Letter 1]

27 Feb 2024

Does radiofrequency ablation procedural data improve the accuracy of identifying atrial fibrillation recurrence?

PONE-D-23-31440R1

Dear Dr. Coplan,

We’re pleased to inform you that your manuscript has been judged scientifically suitable for publication and will be formally accepted for publication once it meets all outstanding technical requirements.

Kind regards,

Ibrahim Marai, MD

Academic Editor

PLOS ONE
---

## [Editor Report · Acceptance letter]

27 Mar 2024

PONE-D-23-31440R1 

PLOS ONE

Dear Dr. Coplan, 

I'm pleased to inform you that your manuscript has been deemed suitable for publication in PLOS ONE. Congratulations! Your manuscript is now being handed over to our production team.

Kind regards, 

on behalf of

Dr. Ibrahim Marai 

Academic Editor

PLOS ONE